# Analysis of Protein Conformational Strains—A Key for New Diagnostic Methods of Human Diseases

**DOI:** 10.3390/ijms21082801

**Published:** 2020-04-17

**Authors:** Andrei Surguchov

**Affiliations:** Department of Neurology, University of Kansas Medical Center, Kansas City, KS 66160, USA; asurguchov@kumc.edu; Tel.: +1-(913)-689-0771

**Keywords:** Parkinson’s disease, multiple system atrophy, α-synuclein, conformational diseases, protein misfolding cyclic amplification

## Abstract

α-Synuclein is a naturally unfolded protein which easily aggregates and forms toxic inclusions and deposits. It is associated with several neurodegenerative diseases, including Parkinson’s disease (PD), dementia with Lewy bodies (DLB), and multiple system atrophy (MSA). These diseases, called synucleinopathies, have overlapping symptoms but require different methods of treatment. There are no reliable approaches for early diagnoses of these diseases, and as a result, the treatment begins late, and the disorders are often misdiagnosed. Recent studies revealed that α-synuclein forms distinctive spatial structures or strains at the early steps of these diseases, which may be used for early diagnosis. One of these early diagnostic methods called PMCA (protein misfolding cyclic amplification) allows identification of the distinct α-synuclein strains specific for different human diseases. The method is successfully used for differential diagnosis of patients with PD and MSA.

## 1. Introduction

The number of diseases due to the amplification of misfolded protein aggregates is quickly growing. Traditional methods used to assay protein amplification, such as immunohistochemistry, enzyme-linked immunosorbent assay, and western blotting, do not have high sensitivity. They require significant amounts of biological samples for reliable analysis. To overcome these drawbacks, more sensitive and robust amplification assays were developed. We briefly describe here ultrasensitive methods assessing the amplification of misfolded protein aggregates. We also outline how efficiently they are for diagnosis of human neurodegenerative diseases.

## 2. Conformational Diseases

Conformational diseases are a large group of disorders that arise when a specific protein changes its conformation, becoming aggregation-prone [1,2]. As a result, the protein accumulates and forms inclusions and deposits in cells and tissues [3]. The aggregation of such conformationally destabilized proteins underlies many neurodegenerative diseases, including such widespread disorders as Alzheimer’s disease (AD) and Parkinson’s disease (PD) [4,5]. Biochemical changes in the brain of patients with neurodegenerative diseases begin many years before the symptoms become evident, and a physician may diagnose the type of the disorder and begin the treatment. By this time, irreversible changes have already occurred, and the treatment becomes difficult. Furthermore, the misdiagnosis of patients with neurodegenerative diseases often happens [6]. Thus, specific diagnostic methods are urgently needed for early identification of the first symptoms of the disease.

## 3. Synucleinopathies

Synucleinopathies are conformational diseases characterized by the excessive accumulation of fibrillary α-synuclein in neurons, nerve fibers or glial cells [7]. α-Synuclein possesses prion-like properties and can easily spread between cells [8]. There are three main types of synucleinopathy: PD, disease dementia with Lewy bodies (DLB), and multiple system atrophy (MSA).

In PD and DLB, fibrillar α-synuclein is deposited in neurons in the form of Lewy bodies and Lewy neurites. In MSA, α-synuclein accumulates in oligodendrocytes forming the glial cytoplasmic inclusion. In addition, in MSA, aggregated α–synuclein is present in neuronal cytoplasmic inclusions, cell processes, and to a lesser extent, in neuronal and glial nuclei [9]. Despite the differences in α-synuclein abnormalities in patients with PD and MSA, these two diseases have overlapping symptoms that are hard to differentiate [6]. This creates a challenge for physicians, as these two diseases require different treatments.

## 4. Protein Misfolding Cyclic Amplification as an Early Diagnostic Method

α-Synuclein aggregation was previously monitored by incubation with thioflavin T (ThT) in shaking tubes. The method has two drawbacks: (1) the procedure is lengthy and took several days or even weeks to measure fibril formation, and (2) the method requires a high concentration of α-synuclein, usually between 300 and 500 µM. Recently an amplification technique called protein misfolding cyclic amplification (PMCA) originally put forward to multiply misfolded prions [10,11,12,13] was modified and adopted for other misfolded proteins. The principle of the method is based on the incubation of a small amount of misfolded protein in the presence of an excess of normal protein. The incubation is conducted in cycles at 37 °C to grow fibrils (Figure 1).

At this step, the misfolded fibrillar protein initiates the conversion of normal protein into misfolded fibrils. The growing chains of misfolded protein are then blasted with ultrasound, breaking it down into smaller chains. This treatment conducted in the presence of Triton X-100 increases the amount of abnormal protein available to cause conversions and prevents precipitation of aggregates. The repetition of the cycles causes further conversions. The method mimics prion replication and has some similarity in principle of polymerase chain reaction (PCR), but it does not use nucleotides. PMCA can be automated and possesses extremely high sensitivity [14]. It may be applied to replicate the misfolded protein from diverse species [15], including the analysis of growing α-synuclein aggregates [16,17].

## 5. PD and MSA Can Be Differentiated at an Early Stage

Although α-synuclein is implicated in several neurodegenerative diseases and the pathological process in these disorders is associated with α-synuclein aggregation, specific conformational strains are formed in each of these diseases. In a recent article Shahnawaz et al. [18] have demonstrated that the PMCA analysis allows discriminating α-synuclein strains from the cerebrospinal fluid of PD and MSA patients. The sensitivity of this analysis is higher than 95%.

Researchers investigated the nature of α-synuclein generated in PMCA reaction by several methods, including fluorescence in the presence of thioflavin T (ThT). ThT binding to amyloids gives strong fluorescence, which allows quantification and monitoring of amyloid fibrils [19]. The research team from several medical centers in Houston (TX), Mayo Clinic in Rochester (MN), USA and Linköping University, (Sweden) analyzed cerebrospinal fluid (CSF) from more than 200 people who had either PD or MSA and compared the results with the data for healthy people [18]. The measurement of fluorescence after ThT binding to α-synuclein aggregates from patients with PD or MSA demonstrated a differential pattern of interaction [18]. These dissimilarities are explained by different conformational strains of α-synuclein [19,20,21,22] present in CSF of patients with PD and MSA. More specifically, samples from PD patients showed higher fluorescence than those from MSA patients confirming that the PMCA technique discriminates between samples isolated from PD and MSA patients. Furthermore, the structural differences between synuclein aggregates from two groups of patients were confirmed by proteinase K digestion and by cryo-electron tomography.

Therefore, different α-synuclein strains cause clinically and pathologically distinct diseases. Importantly, different α-synuclein strains target distinct cell types within the brain, and strain-specific differences are maintained after serial passaging, implying that α-synuclein propagates via prion-like conformational templating [23]. The conformation adopted by α-synuclein assemblies defines their ability to amplify, propagate in the brain in vivo, and causes different pathologies. Growing evidence indicates that different strains may specify distinct phenotypes not only in synucleinopathies but also in other neurodegenerative diseases, for example, caused by accumulation of β-amyloid and tau [24,25]. The PMCA method can also be used for testing brain homogenates, opening an opportunity for laboratory investigation of protein propagation in a biologic complex matrix [26].

Recent progress in the PMCA method and other techniques of protein analysis is a first step to precision medicine, which hopefully will allow differentiating not only each type of synucleinopathies and other conformational diseases but even determine the type of protein conformation-specific for each individual patient [27]. These recent developments raise hope that such methods of early diagnostics may become available in the future for such diseases as Alzheimer’s disease, progressive supranuclear palsy, frontotemporal dementia, and Parkinsonism linked to chromosome 17. Another important translational application of PMCA is a possibility to assess aggregation-inhibitory molecules with potential therapeutic applications.

## 6. Comparison of PMCA with Real-Time Quaking-Induced Conversion (RT-QuIC) Assay

After PMCA method was developed [10] and modified by the same research team for prions [11,12,13,14] and for α-synuclein and other proteins [15,16,17], another assay allowing amplification of prions or prion-like proteins was put forward. The method is called a real-time quaking-induced conversion (RT-QuIC) assay [28]. For both assays, a control and experimental sample are mixed with amplification substrate (brain extract, biological fluid, recombinant protein, etc.). The mixture is then subjected to cyclic amplification over a number of cycles of sonication/degradation for PMCA or shaking for RT-QuIC at an appropriate temperature. During this treatment, protein amplification products accumulate in the test tubes. These products are analyzed by Western blotting for PMCA and by fluorescent emission reading for RT-QuIC method. Generally speaking, equipment and amplification cycling methods can definitely be developed in a typical biochemical laboratory.

Each of the two methods has its advantages and disadvantages and may be sensitive to different inhibitors. For example, members of glycoprotein family mucins are potent inhibitors of RT-QuIC reaction [29]. This is a serious limitation for the analysis of saliva where mucins concentration is very high. To circumvent the inhibition, Davenport and coauthors developed a modified PMCA with high detection sensitivity [29]. In another study endogenous polar lipids inhibited RT-QuIC reaction in brain samples. The removal of polar lipids by alcohol-based extraction treatment enabled detection and restored a linear dilution response [30]. Other details of experimental procedures and comparison of PMCA and RT-QuIC methods can be found in recent publications [31,32].

## Figures and Tables

**Figure 1 ijms-21-02801-f001:**
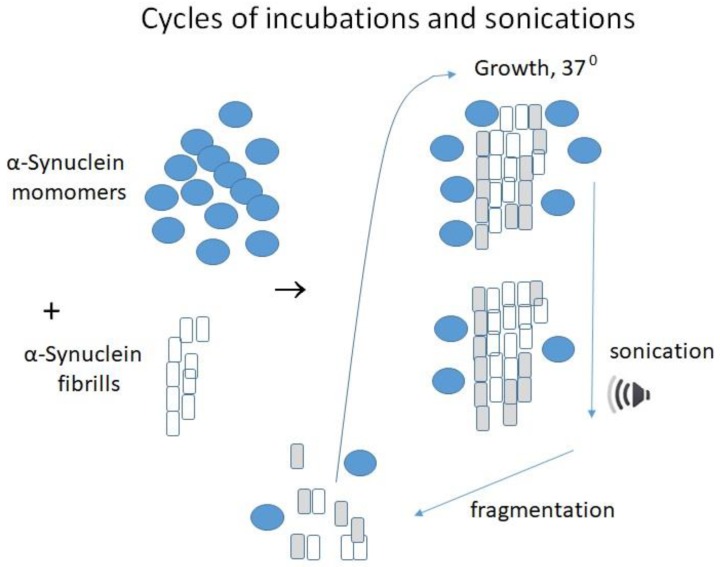
The PMCA is a cyclic reaction, combining the growing and multiplication of the template units. A small amount of fibrillar α-synuclein is incubated with an excess of monomeric α-synuclein. After some conversion occurs, the mixture is blasted with ultrasound, breaking it down into smaller pieces. As a result of this step, the amount of fibrillar protein available to cause further conversions is rapidly increased. By repeating the cycle, the mass of monomeric α-synuclein is quickly changed into the fibrillar protein. The method may be automated, causing a significant increase in the efficiency of amplification.

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
