# Peer review of "Analysis of Protein Conformational Strains—A Key for New Diagnostic Methods of Human Diseases"

_ijms, 2020, doi:10.3390/ijms21082801_

Round 1
Reviewer 1 Report
This is an interesting short review focused on a novel experimental approach in the study of biomarkers. My first concern is on the fact that this is a review not a communication of results. Taking into account this premise, the topic chosen by the author is novel and really promising in the field of synucleinopathies and, potentially, in all the diseases that are linked to protein aggregation. In the case that International Journal of Molecular Sciences would admit short reviews instead of Communication of results, in my opinion, some minor points have to be addressed before acceptance for publication.
- The author focuses on PMCA assay and discuss principles, very recent applications and future perspectives. In my opinion, this review lacks the comparison with other recent approaches that have the same goals and that led to relevant results. A poor mention to this issue is reported at page 3, line 1: “Recent progress in the PMCA method and other techniques of protein analysis is a first step to precision medicine,….”. Please expand this issue and cite which are these “other techniques”, precisely because this is a review. In addition, the very recent data coming from those approaches could be briefly included for discussing if they agree with those coming from PMCA or not.
- The order of paragraphs is inexplicable: paragraph 1 is titled “introduction” but, indeed, it does not exist, no sentences are included for that. This is not a research paper in which we put introduction, methods…. However, also a review might include an introduction, but the author could have to write some introductive sentences there.
- Please check for typos. For an example: “forms” instead of “form” in the first line of abstract; Page 2. Lines 6 and 7, “1) the...” and “2) the..) instead of “1) The...” and “2) The..
Author Response
Reviewer 1
We would like to thank reviewers for their critique and suggestions. We attached two versions of the corrected manuscript, in one of which corrections are marked by red color.
Comment 1: ”In the case that the International Journal of Molecular Sciences would admit short reviews instead of Communication of results…”
Response: I agree and believe that the Editorial Board of the International Journal of Molecular Sciences will assign the manuscript to the section, which is the most appropriate. Initially, I suggested submitting it as Communication because a part of the manuscript is devoted to the application of PMCA method for the diagnosis of PD and MSA. Now I extended the manuscript per the recommendation of the reviewers. I would gladly follow the instructions of the Editorial board if the section of the journal were different from Communication.
Comment 2:” this review lacks the comparison with other recent approaches that have the same goals and that led to relevant results. A poor mention to this issue is reported on page 3, line 1: “Recent progress in the PMCA method and other techniques of protein analysis is the first step to precision medicine,….”. Please expand this issue and cite which are these “other techniques”, precisely because this is a review. In addition, the very recent data coming from those approaches could be briefly included for discussing if they agree with those coming from PMCA or not.
Response: As an expansion of this issue, we added a new chapter at the end of the manuscript called “Comparison of PMCA with real-time quaking-induced conversion (RT-QuIC) assay. We also added corresponding references in the bibliography with the description of other techniques (Ref. 28) and the comparison of the techniques (Ref. 29-32).
Comment 3:” The order of paragraphs is inexplicable: paragraph 1 is titled “introduction,” but, indeed, it does not exist, no sentences are included for that...author could have to write some introductive sentences there”.
Response. Thank you very much for this suggestion. We added several sentences into Introduction, beginning with “The number of diseases due to the amplification of prions, prion-like proteins, various forms of amyloidosis. etc. is growing.”
Comment 4. Please check for typos.
Thank you, typos are corrected.
Reviewer 2 Report
The manuscript prepared by Andrei Surguchov is a very brief summary about selected methods that promise new diagnostic tools of human neurodegenerative diseases.
The topic is of great importance and broad interest, and the reader could find some interesting information, mostly about a method called protein misfolding cyclic amplification (PMCA), which was recently successfully tested by several research groups.
The article is in general well written, but its narrow form fails to give a concise overview of the described methods. This short commentary lacks objective discussion about all benefits and drawbacks of the PMCA method, although the reader would expect that while looking at the title. Therefore, I suggest the author to extend the manuscript in order to provide the reader with more detailed picture of both the PMCA method and other methods (or combined approaches) that have a potential to revolutionize diagnostics of neurodegenerative diseases. In its present form the manuscript seems to be too sketchy.
Author Response
Reviewer 2
The article is in general well written, but its narrow form fails to give a concise overview of the described methods. This short commentary lacks objective discussion about all benefits and drawbacks of the PMCA method, although the reader would expect that while looking at the title. Therefore, I suggest the author to extend the manuscript in order to provide the reader with a more detailed picture of both the PMCA method and other methods (or combined approaches) that have the potential to revolutionize diagnostics of neurodegenerative diseases. In its present form the manuscript seems to be too sketchy.
Response. In response to this comment, we added a new part at the end of the manuscript entitled “Comparison of PMCA with real-time quaking-induced conversion (RT-QuIC) assay”. We also added several references (28-32) with a more detailed description of the methods and their comparison.
Thank you very much for your critiques and suggestions
Round 2
Reviewer 2 Report
I would like to thank the author for the fragments added to the manuscript, although I have an impression that the topic could be presented more precisely and in a bit broader context.